# Extension of Solid Solubility and Structural Evolution in Nano-Structured Cu-Cr Solid Solution Induced by High-Energy Milling

**DOI:** 10.3390/ma13235532

**Published:** 2020-12-04

**Authors:** Liyuan Shan, Xueliang Wang, Yaping Wang

**Affiliations:** 1MOE Key Laboratory for Nonequilibrium Synthesis and Modulation of Condensed Matter, Xi’an Jiaotong University, Xi’an 710049, China; shanliyuan7981@163.com; 2MOE Key Laboratory of Thermo-Fluid Science and Engineering, Xi’an Jiaotong University, Xi’an 710049, China; xlwang082@mail.xjtu.edu.cn; 3State Key Laboratory for Mechanical Behavior of Materials, Xi’an Jiaotong University, Xi’an 710049, China

**Keywords:** Cu-Cr system, mechanical alloying, solid solubility extension, structural evolution, thermodynamic

## Abstract

In Cu-Cr alloys, the strengthening effects of Cr are severely limited due to the relatively low Cr solid solubility in Cu matrix. In addition, apart from the dissolved Cr, it should be noted that high proportion of Cr in Cu matrix work as the second phase dispersion strengthening. Therefore, it is of great significance to extend the Cr solid solubility and decrease the size of the undissolved Cr phase to nano-structure. In this work, the nano-sized Cu-5 wt.% Cr solid solution was achieved through high energy ball milling (HEBM) only for 12 h. The Cr solubility of ~1.15 at.% was quantitatively calculated based on XRD patterns, which means supersaturated solid solution was realized. Except for the dissolved Cr, the undissolved Cr phase was with nano-sized work as the second phase. Upon milling of the Cu-Cr powders with coarse grains, the crystallite sizes and grain sizes are found to decrease with the milling time, and remain almost unchanged at a steady-state with continued milling. In addition, it was found that the stored energy induced by dislocation density increment and grain size refinement would be high enough to overcome the thermodynamic barrier for the formation of solid solution.

## 1. Introduction

Cu-Cr alloys are widely used in the electrical industry, such as contact materials and lead frames in integrated circuits, due to the well mechanical properties and high electrical conductivity [1,2,3]. The excellent mechanical properties are mainly produced by the formation of fine tiny Cr precipitates in Cu matrix through aging treatment, known as age or precipitation hardening [4]. However, the effect of precipitating strengthening of Cr particles is limited due to the limited maximum solubility at eutectic temperature [5]. In addition, for Cu-Cr alloys with high Cr content (>5 wt.%), the undissolved Cr phase possesses the majority proportion compared with dissolved Cr, and the coarse undissolved Cr particles may deteriorate the mechanical properties [6,7]. Therefore, it is of great significance to extend the Cr solid solubility in Cu matrix and achieve nano-sized undissolved Cr particles in Cu matrix, so that the precipitation hardening and dispersion strengthening can be combined. 

In recent years, to extend the Cr solubility in Cu matrix, technology methods such as rapid solidification (RS) [8], severe plastic deformation (SPD) [9], high energy ball milling (HEBM) have been introduced and improved [10]. The common feature of the aforementioned techniques is the possibility to drastically extend the solid solubility level of nearly immiscible elements in alloys by non-equilibrium growth conditions [11,12], offering a promising route to attain high-performance composites. However, in the case of the undissolved Cr phase, RS method can only achieve micron-sized (0.1 μm) residual Cr particles, which cannot strengthen Cu matrix effectively. Mechanical alloying (MA) using high energy ball mill is not only a promising method to extend the solid solubility, but also a practical approach to achieve the homogeneous distribution of nano-structural materials from starting blended powders [10]. In addition, during the milling process, a large number of structural defects occur, such as vacancies, dislocations and stacking faults. Additionally, high energy can transfer easily to the powders due to the increase in the grain boundary volume. Under these conditions, the diffusivity of the solute atoms increase, and solid solubility extension can be achieved, namely supersaturated solid solution. 

Sheibani and Fang et al. have prepared Cu-Cr alloying powders by MA treatment for 50 h or a longer time and supersaturated Cu-based solid solution (<1 at.% in case of Cu–Cr) has been formed [13,14,15]. Although the results showed that the MA method can increase Cr solubility in Cu matrix, and Cu-Cr bulk fabricated from the powders with MA treatment possessed high mechanical properties (1.6 Gpa) [16], there was no quantitative calculation of Cr solubility as a function of milling time and the strengthening effect of residual nano-sized Cr phase. A quantitative understanding of the formation process of supersaturated solid solution could help to design high-strength and high-conductivity nano-structural Cu-Cr alloy. More importantly, it should be mentioned that amount of impurities may get into the milled powder and contaminate it during MA treatment for such a long time (>50 h) [11,17]. In some cases, contaminations, such as Fe element, cannot be ruled out in MA materials and might significantly influence the measured properties, especially when the container is produced by stainless-steel [10].

In the present study, Cu-5 wt.% Cr blended powders were MA treated for 12 h, which is shorter than that of other studies. XRD patterns are employed to study the effect of milling time on the microstructure evolution, microstructural parameters such as the crystallize size, lattice parameter and dislocation density. The Cr solid solubility of ~1.15 at.% was quantitatively calculated based on XRD patterns, which means supersaturated solid solution was realized. Except for the dissolved Cr, the undissolved Cr phase was with nano-sized work as the second phase. In addition, thermodynamic analysis of the driving force and mechanism for the formation of supersaturated solid solution in the Cu-Cr system was conducted. The morphology and microstructure were characterized by scanning electron microscopy (SEM) and transmission electron microscopy (TEM) equipment. 

## 2. Experimental Procedure

This work aims to extend the solid solubility of Cr and achieve nano-sized undissolved Cr particles in Cu matrix using the HEBM method. Figure 1 shows the schematic illustration of morphology changes of Cu-Cr powders during MA process, which mainly includes the fracture and welding stage. Commercial pure Cu powders and Cr powders (purity >99.5% and particle size smaller than 75 μm) were used as the raw materials. A total of 10 g of blend compositions with 5 wt.% Cr were subjected to high-energy ball milling for 12 h in stainless-steel grinding media by a Spex-8000 mill. The ball to powder weight ratio was set as 20:1, and the diameter of the milling ball was in the range of 5–8 mm. During the milling process, 2 wt.% ethanol was added as a process control agent to alleviate the aggregation of powders and realize a homogeneous supersaturated solid solution [13]. The effects of HEBM on the microstructure evolution and solid solubility extension of Cu-Cr alloy powders were investigated through different milling times (2, 4, 8, 10 and 12 h). To further evaluate the effect of the milling process on the dissolution ability of Cr in Cu matrix, Cu-5 wt.% Cr powders with MA treatment for 12 h was aging treated at 550 °C for 3 h.

X-ray diffraction (XRD, Bruker D8 ADVANCE, Leipzig, Germany) with Cu Kα radiation (λ = 0.15406 nm) was utilized to determine the structural evolution of Cu-Cr powders. The patterns were recorded in a 2θ scan ranging from 20° to 100°. The crystallite size (*d*) powder was determined according to the Williamson–Hall plot [18]. The line broadening due to the instrument was calculated from Warren’s methods [19,20]. The internal strain (ε) was calculated according to Equation (1) [21,22],
(1)$$\varepsilon =\frac{\beta }{4\;\tan\theta }$$

In this work, the dislocation density (ρ) can be estimated through Equation (2) based on the XRD data [1,21,22,23],
(2)$$\rho =\frac{k\varepsilon }{b^{2}}$$
where *k* is a geometrical constant with a value of 16.1 for face-centered cubic materials, *b* is the Burgers vector of Cu with the value of 0.256 nm [21,24]. Meanwhile, the lattice parameter of Cu was also evaluated based on the XRD patterns.

The morphology and microstructure of Cu-Cr alloying powders were investigated using SEM (JEOL JSM-7000F, Tokyo, Japan) technique. TEM images of the samples were observed with a JEOL JEM-2100 TEM instrument operated at 120 keV with the samples ultrasonicated in ethanol for 30 min and dispersed on carbon film supported on copper grids (200 mesh). The average particle size (*D*) was estimated from the SEM images (the average value is obtained from more than 100 measurements).

## 3. Thermodynamic Theory

The thermodynamic theory was utilized to predict the stability of the Cu-Cr system. In the standard state, the Gibbs free energy (Δ*G*) for forming ideal A(B) solid solution from elemental powders of pure A and B can be defined as [25]: (3)ΔG=ΔH−T×ΔS
where Δ*H* and Δ*S* are the formations of enthalpy and entropy of solid solution, respectively. *T* is the absolute temperature at which solid solution formation is realized. Considering the rise in temperature during the milling process, the temperature of 473 K is adopted to calculate Δ*G*. Specifically, Δ*S* can be presented as follows:(4)$$\Delta S=-R\left(x_{A}\times lnx_{A}+x_{B}\times lnx_{B}\right)$$
where *R* is the universal gas constant (8.31 J/(mol·K), *x*_A_ and *x*_B_ are the mole fractions of elementals A and B, respectively, *x*_A_
*+ x*_B_ = 1.

Δ*H* consists of three terms based on the semi-empirical model, which can be evaluated by the following equation [25]: (5)$$\Delta H=\Delta H_{C}+\Delta H_{E}+\Delta H_{S}$$
where Δ*H*_C_ is the enthalpy change resulting from chemical contribution, Δ*H*_E_ represents elastic mismatch energy associating with size mismatch between solid solution and starting materials. Δ*H*_S_ is the structural contribution enthalpy due to the crystal structure difference in the mixture atoms. For the Cu-Cr system, the effect of Δ*H*_S_ is so small that it can be neglected in the calculation process [26]. 

Specifically, Δ*H*_C_ for binary alloy can be characterized by Equation (6) [27]:(6)$$\Delta H_{C}=2P\times \frac{x_{A}\times V_{A}{}^{\frac{2}{3}}\times V_{B}{}^{\frac{2}{3}}/\left(x_{A}\times V_{A}{}^{\frac{2}{3}}+x_{B}\times V_{B}{}^{\frac{2}{3}}\right)}{n_{A}{}^{\frac{-1}{3}}+n_{B}{}^{\frac{-1}{3}}}\times [-\varphi^{2}+\frac{Q}{P}\left(n_{A}{}^{\frac{1}{3}}-n_{B}{}^{\frac{1}{3}}{)}^{2}\right]$$
where *P* and *Q* are the empirical parameters related to elementals A and B. In numerical calculation, *P* is adopted as 14.1, and *Q*/*P* is determined as 9.4 for the Cu-Cr system [27]. *V*, φ and *n* are the mole volume, electron chemical potential and electronic density, respectively [28,29]. 

Δ*H_E_* arises from elastic contribution would be given by Equation (7):(7)$$\Delta H_{E}=x_{A}\times x_{B}\left(x_{A}\times \Delta H_{A}{}^{B}+x_{B}\times \Delta H_{B}{}^{A}\right)$$
where Δ*H*_A_^B^ and Δ*H*_B_^A^ are the elastic energy contributed by A dissolving in B and B dissolving in A, respectively. In addition, Δ*H*_A_^B^ and Δ*H*_B_^A^ would be estimated as follows: (8)$$\Delta H_{A}{}^{B}={\frac{2K_{A}\times G_{B}{}^{\;}{\left(V_{A}-V_{B}\right)}^{2}}{3K_{A}\times V_{A}+4G_{B}\times V_{A}}}_{\;}$$
(9)$$\Delta H_{B}{}^{A}={\frac{2K_{B}\times G_{A}{}^{\;}{\left(V_{B}-V_{A}\right)}^{2}}{3K_{B}\times V_{B}+4G_{A}\times V_{B}}}_{\;}$$
where *K* and *G* denote the bulk modulus and shear modulus, respectively.

Table 1 gives the input parameters in Equations (3)–(9). Accordingly, the formation enthalpy, entropy and Gibbs free energy of the Cu-Cr system at the temperature of 473 K can be calculated, as shown in Figure 2. It is easily deduced that the contribution of △*H*_C_ is main to form Cu-Cr solid solution. This may be due to the obvious difference of bonding energy between solid solution and the starting mixture. However, the contribution of elastic energy can be neglected due to the near zero value of Δ*H*_E_, which can be confirmed by the small difference in molar volume between Cu and Cr, as shown in Table 1. Moreover, it should be noted that the value of Gibbs free energy is positive in alloy materials, which means that external energy should be introduced to overcome the energy barrier to achieve stable Cu-Cr supersaturated solid solution. 

## 4. Results and Discussion

### 4.1. Microstructure Evolution

XRD analysis was performed to evaluate the effect of milling time on the structural evolution of Cu-Cr powder, the pattern of raw Cu-5 wt.% Cr mixtures are presented as a reference. In the XRD patterns of all samples, the strongest face-centered cubic (fcc) diffraction peak of Cu (111) appears at 2θ = 43.297°. In addition, the strongest body-centered cubic (bcc) Cr (110) peak in the standard pattern appears at 2θ = 44.392°. According to the studies of Bachmaier and Zhao, in which the lattice parameter changes for Cu, calculated from the (200, 220, 311) and (111, 222) peaks, are quite similar [27,28]. Therefore, the specific XRD patterns are analyzed based on Cu (111) peak within the range of 2θ = {41.5, 45.5} in this work, as shown in Figure 3. No significant signs of Cr peaks can be observed in the XRD pattern, this can be attributed to the lower volume of Cr phase.

In Figure 3a, it can be clearly visible that the fcc Cu (111) peaks become wider with the increase of milling time. The Cu peaks are broadened by increased milling time since the continuous deformation of powder particles during the milling process results in crystallite refinement and an increase in lattice microstrain, which is consistent with the research of Sheibani [13,30]. In addition, the peak intensity reduction and shift of diffraction angle can be observed, which display the typical behaviors of MA treatment powders. Interestingly, the relative intensity and width of Cu (111) peaks exhibit no significant change with the milling time of 8, 10 and 12 h, indicating that the diffusivity of Cr in Cu substrate becomes slow after MA treatment for 8 h. 

Figure 3b shows the XRD patterns of MA treatment for 12 h to Cu-5 wt.% Cr powders and then followed by aging treatment. In comparison, the Cu (111) peak of the sample with aging treatment becomes sharper, as well as the intensity is higher. It seems like the alloying powders after aging treatment transformed into the mixture of the Cu phase and Cr phase. The difference in XRD patterns results from the dissolved Cr atoms precipitated out of Cu substrate during the subsequent heat treatment. 

To further evaluate the effect of milling time on Cu-Cr alloyed powders, the crystallize size, lattice parameter and dislocation density of Cu in all samples were analyzed based on the XRD data. As shown in Table 2, the crystallize size decrease gradually with the increase of milling time. In addition, Cu lattice parameter has a different rising due to the dissolubility of Cr atoms. These results are in excellent agreement with the previous research, in which Sheibani believed that the only evidence of solid solubility extension was lattice parameter changes [13]. Deserved to be mentioned, up to 8 h of milling time, the rapid changes of crystallize size and lattice parameter can be noticed, after that a slower change trend is observed, which could be also confirmed with the small change of diffraction peak intensity and peak width as previously mentioned. Therefore, it could be concluded that in the Cu-Cr powders with coarse grains, the crystallite sizes are found to decrease gradually with the milling time, and remain small changes with continued milling. This phenomenon can be explained by the following reasons:

At the beginning of MA process, the ductile Cu matrix was subjected to the ball collisions, leading to the improvement of surface activity. The brittle Cr phase fractured and size decreased more quickly, being easy to adhere onto the soft Cu surface and transfer into Cu lattice. However, under repeated collisions, the surface layer of Cu particles got hardened due to the dissolution of solute atoms, so that the diffusion of Cr and variation of crystallize size are going to be difficult. The steady-state value was proposed and needs to be verified further. 

### 4.2. Morphology of Cu-Cr Alloying Powders

SEM morphologies concerning milling time are shown in Figure 4. As marked in Figure 4a, raw Cu powders have the typical dendritic appearance, coarse Cr powders are used for the strengthening phase. The mixed powders transform to a layer-shaped structure due to the excellent ductility of Cu in the initial stage of MA process, tending to be easily cold welding and appear as large layer [14]. After that, the fracture occurs more frequently than cold welding, so that the layered structure gradually refined, as presented in Figure 4c. In addition, the increase of surface energy and the defect can make the refined Cr adhere easily onto Cu particles. By comparing Figure 4e,f, there is no big difference in particle size and morphology. However, the coarsening of powders cannot be avoided even milling with 12 h due to the existence of cold welding, as displayed in Figure 4f. During the long-time of milling, the harder Cr particles not only act as the abrasive particles but also can dissolve in the softer Cu particles due to the high density of structural defects.

The average particle size changes as a function of milling time are shown in Figure 5. It can be observed that the average particle size of sample milled with 2 h is a little larger than that of raw materials. In addition, MA treatment for 12 h to sample possess the minimum particle size. This result is consistent with that of Figure 4, in which the particle size becomes coarse firstly and then decreases tending to remain unchanged with continued milling. The mentioned phenomena also indicate that the solid solubility increase trend becomes slower after 8 h of milling, which will be estimated in the next part.

### 4.3. Effect of Milling Time on the Solid Solubility

The solid solubility level is related to the lattice parameter changes of the matrix obtained either for equilibrium or nonequilibrium processing techniques [10]. More importantly, the structure factors for alloyed phases produced by method of rapidly solidified, irradiated, sputtered and MA have been found to be very similar [31]. As mentioned previously, Sheiban believed that the only evidence of solid solubility extension was lattice parameter changes [12]. In this regard, the solid solubility of Cr in Cu matrix obtained by MA treatment can be determined by Equation (10) [4]:(10)c=δa/0.00026
where *c* is the Cr content in MA treated Cu-Cr alloy, at.%, δ*a* denotes the lattice parameter variation due to the dissolubility of Cr atoms, nm. The value 0.00026 is the factor, nm/at.%, which was established for the extended solid solubility obtained by chill block melt spinning [4,32].

Figure 6 displays the changes in Cr solubility and Cu internal strain concerning milling time. With the increase of milling time, Cr solubility and the internal strain of Cu both increase rapidly, especially at the 4 h of milling time for Cr solubility. The increase of Cu internal strain is mainly due to the refinement of crystallize size, as displayed in Table 2. Crystalize size decreased rapidly up to 4 h of milling time, and slowly decreased from there. However, the lattice parameter increased in a steady-stage during the MA process. In the case of the internal strain, the increase is slow for the entire time of milling, while the Cr solubility increased sharply within the initial 4 h of milling. The slower increase rate after 4 h would be attributed to the diffusion of Cr becoming difficult. In addition, it can be noted that the Cr solubility (1.07 at.%, 1.15 at.%) reach to the usual saturation (0.89 at.%) with milling time more than 8 h. The supersaturated Cu-Cr solid solution powder is prepared successfully with MA treatment for more than 8 h in this work. According to the trace of Cr solubility curve, the Cr solubility tends to increase very slowly or reach a constant with the increase in milling time, as well as the crystallize size.

### 4.4. Gibbs Free Energy Changes of Cu-Cr Alloying Powders

The Gibbs free energy in the Cu-Cr system will change due to the formation of solid solution. As mentioned in Table 2, with the increase of milling time, the typical crystallite size refinement and dislocation density increase can be observed. On one hand, the increment of dislocation density during the milling process would contribute to the total Gibbs free energy (Δ*G*_t_). The energy change due to the dislocation density increment, Δ*G*_d_, can be calculated by Equation (11) [13]:(11)$$\Delta G_{d}=\zeta \rho V_{m}$$
where ρ and *V*_m_ are the dislocation density and molar volume, respectively. The molar volume of Cu is 7.1 cm^3^/mol [13]. ζ, the dislocation elastic energy per unit length of dislocation lines, can be calculated using the following equation [33]:(12)$$\zeta =\frac{Gb^{2}{}_{\;}}{4\pi }\times ln\left(\frac{R_{e}}{b}\right)$$
where *G* = 4.8 × 10^10^ N/m^2^ is the shear modulus for Cu [22], *b* is the burgers vector. *R*_e_ is the outer cutoff radius of dislocation, which can be taken as crystallite size in nanocrystalline materials [13,32]. 

On the other hand, the Gibbs free energy in the Cu-Cr system also increases with the refinement of crystallite size, which determined by Equation (13):(13)$$\Delta G_{c}=\gamma \left(\frac{A}{V}\right)V_{m}$$
where γ is the grain boundary energy with the value of 625 mJ/m^2^ for Cu, *A*/*V* and *V*_m_ are the surface volume ratio and molar volume, respectively. To determine Δ*G*_c_*,* spherical crystallite was considered [34]. 

Figure 7 displays the Gibbs free energy increment due to the variation of crystallite size and dislocation density. It can be seen that the increment of Gibbs free energy resulting from crystallite size and dislocation density changes are ~1.45 and 0.99 kJ/mol with MA treatment for 12 h, respectively. It can be visible that the contribution of dislocation density is higher than that of crystallite size. The total Gibbs free energy, Δ*G*_t_, also was calculated based on Δ*G*_c_ and Δ*G*_d_. Therefore, the maximum Δ*G*_t_ with milling time for 12 h would be 2.44 kJ/mol. By comparing 2.44 kJ/mol with the value in Δ*G* trace in Figure 1, it can be concluded that the energy deriving from MA process would be high enough to form the Cu-Cr supersaturated solid solution. In other words, the stored energy obtained by the variation of crystallite size and dislocation density can extend the Cr solid solubility in Cu matrix.

However, the thermodynamic barrier is about 2.62 kJ/mol in Figure 1 while forming the same content Cr (1.15 at.%, MA treatment for 12 h) supersaturated solid solution, which is a little higher than 2.44 kJ/mol. The small difference in these two values may be due to only mean crystallite size was employed to estimate the energy. The smaller size crystallite was neglected which presents a lognormal distribution [34]. In addition, although a large number of structural defects occur during MA process, such as vacancies, dislocations and stacking faults, only dislocations were considered to estimate the energy increase.

### 4.5. HRTEM Characterization

In addition to the XRD technique, TEM equipment was also employed to study the microstructure of the sample subjected to milling for 12 h. Figure 8a shows the typical bright-field image, numerous nanograins (~100 nm) can be visible in the bright-field image, which is agreed well with that estimated by the Debye Scherrer equation. The corresponding selected area electron diffraction (SAED) analysis is inserted in the bottom-right corner, in which Debye–Scherrer rings exhibit two kinds of diffraction rings. As marked, (200), (220) and (311) diffraction rings belong to the fcc Cu phase, and an additional (110) ring pattern can be attributed to the bcc Cr phase. The existing Cr ring pattern demonstrates that there are residual nano Cr particles or Cr-rich phase except for the dissolved Cr in Cu matrix, which cannot be observed in XRD patterns due to the small amount. It should be mentioned that the residual nano-sized Cr phase also can act as secondary strengthening particles by dispersing uniform in Cu matrix. This phenomenon is comparable with the calculated solid solubility result, in which the solid solubility value is ~1.15 at.% after MA treatment for 12 h. More importantly, the lattice parameter of fcc Cu (200), Cu (220) and Cu (311) ring patterns in Figure 8a are 0.36199, 0.36212 and 0.36215 nm, respectively, which are larger than that of pure Cu. The increase in lattice parameter can be attributed to Cr atoms dissolve in Cu matrix, which has been estimated before in the analysis of XRD patterns. There is a little difference in the result of lattice parameters determined from XRD and TEM since both methods present different sensitivities concerning small misorientations of the crystal lattice [17]. Moreover, from the SEM and TEM images, it can be observed that samples subjected to milling for 12 h have good homogeneity. 

To further understand the atomic structure of supersaturated solid solution Cu-Cr powders, the high-resolution TEM (HRTEM) image was taken, as presented in Figure 8b. It can be observed that several Cu crystallites are distributed around a residual nano-sized Cr crystallite based on the difference of lattice fringe. In addition, Cu grains have evident twins and deformed zone (as indicated by double arrows), which is different from Cr grains. The interface of Cu and Cr (labeled with white dashed line) is ~2–3 nm, which is visible clearly. However, the interface at other regions is ambiguous, this may be caused by Cr atoms dissolving in Cu lattice or by severe deformation after MA treatment, so that two or more grains overlapping. 

## 5. Conclusions

Supersaturated Cu-Cr solid solution has been formed by the MA method. This could be noticed from the XRD data and the variation in the lattice parameter of Cu. The changes of lattice parameter are believed the only evidence of solubility extension, and the quantitative calculated results show that the Cr solubility is ~1.15 at.% for the 12 h milling powders. Morphology and particle size have been changed during MA, in the early stage processing, especially up to 4 h, the average particle size decreased rapidly with increasing milling and the internal strain accumulation. In the case of thermodynamic changes, the Gibbs free energy induced by crystallite size decrease and dislocation density increment in the Cu-Cr alloy system would be high enough to increase the Cr solubility. 

## Figures and Tables

**Figure 1 materials-13-05532-f001:**
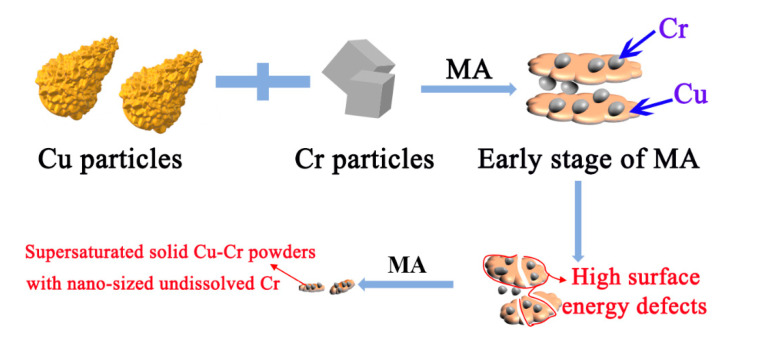
Schematic presentation of morphology changes during mechanical alloying (MA) process average particle size.

**Figure 2 materials-13-05532-f002:**
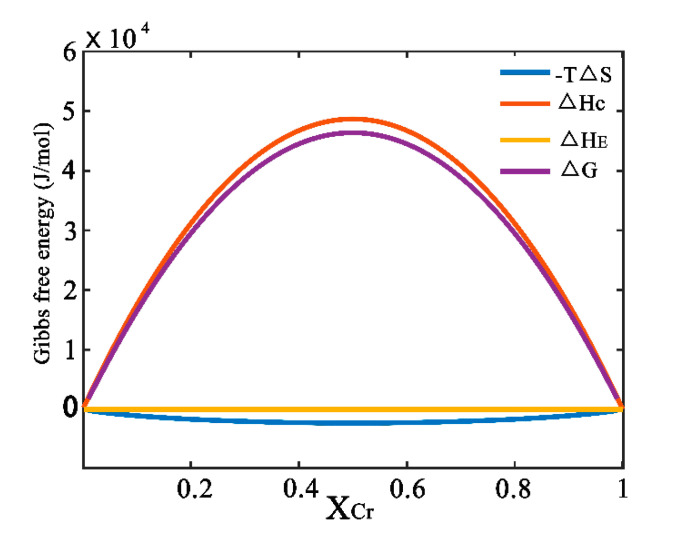
Enthalpy, entropy and Gibbs free energy of the Cu-Cr system at the temperature of 473 K.

**Figure 3 materials-13-05532-f003:**
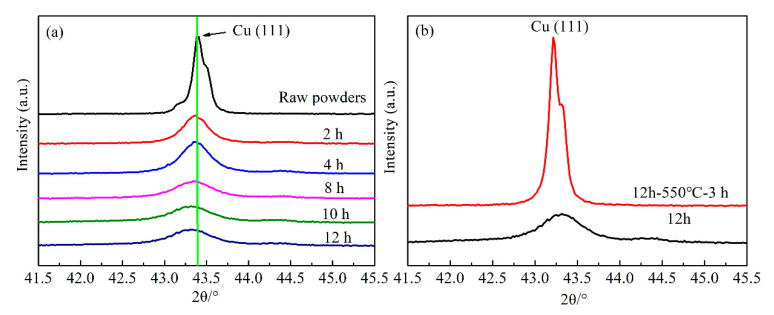
XRD patterns of Cu-Cr powders: (**a**) pattern changes as a function of milling time; (**b**) comparison of Cu (111) peak in the final samples before and after aging treatment.

**Figure 4 materials-13-05532-f004:**
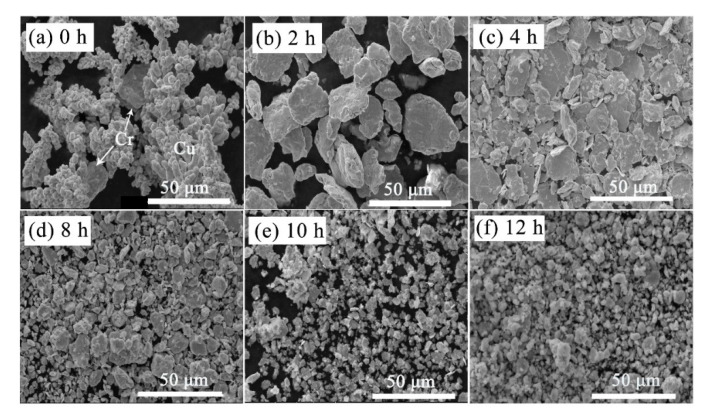
Morphology changes of Cu-Cr powders with the milling time. (**a**) 0 h, (**b**) 2 h, (**c**) 4 h, (**d**) 8 h, (**e**) 10 h, (**f**) 12 h.

**Figure 5 materials-13-05532-f005:**
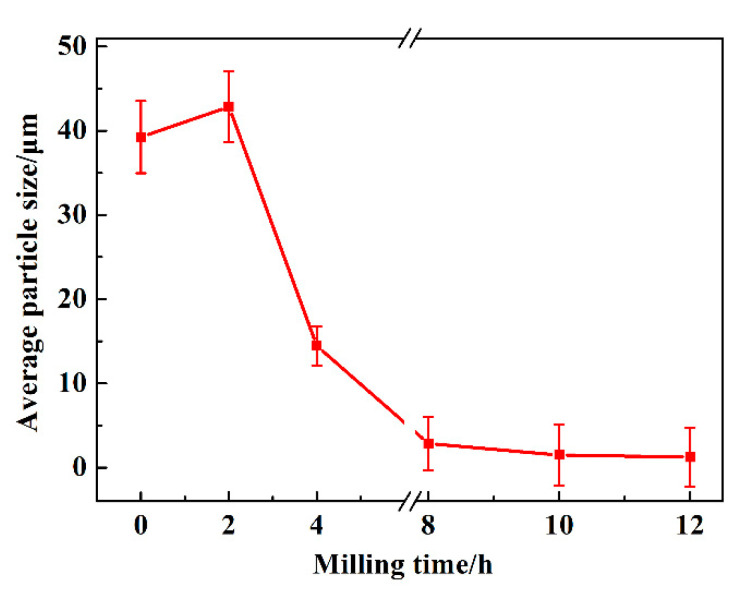
The average particle size of Cu-Cr alloying powders as a function of milling time.

**Figure 6 materials-13-05532-f006:**
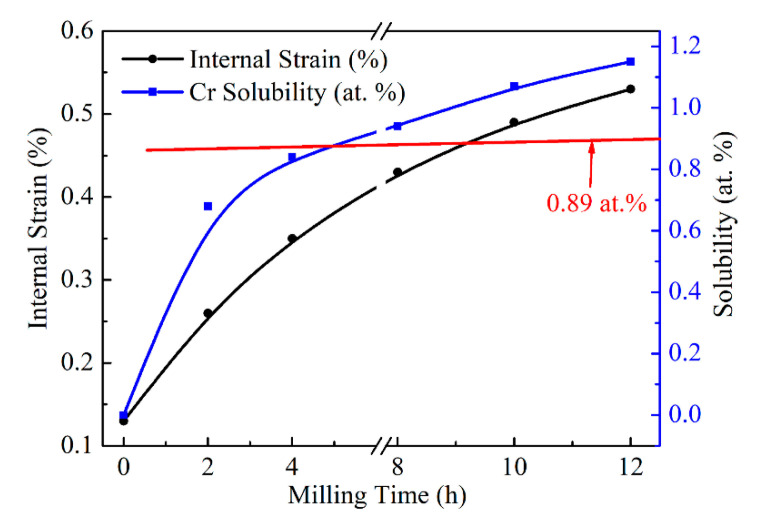
Effect of milling time on Cu internal strain and Cr solid solubility.

**Figure 7 materials-13-05532-f007:**
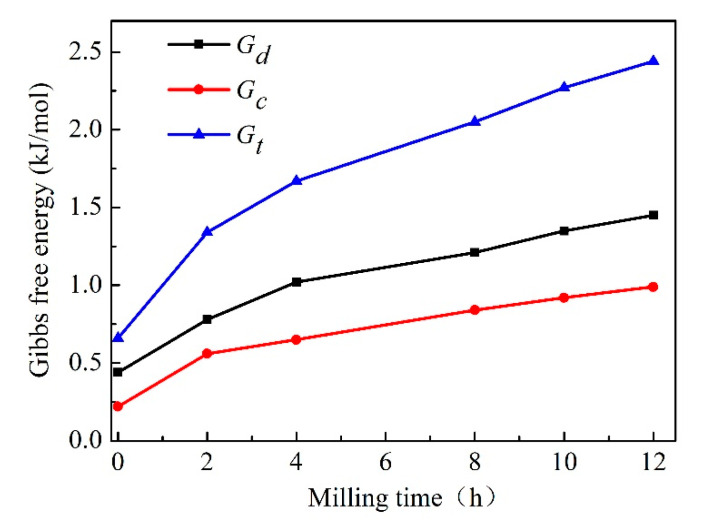
Effect of Cu crystallite size and dislocation density on Gibbs free energy.

**Figure 8 materials-13-05532-f008:**
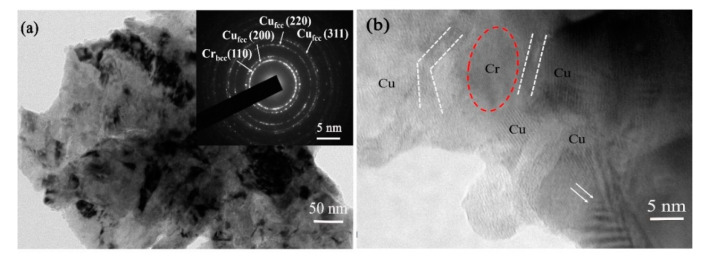
The microstructure of Cu-Cr alloying powders subjected to MA for 12 h: (**a**) bright-field image (the inset bottom-right is SAED pattern); (**b**) typical HRTEM image.

**Table 1 materials-13-05532-t001:** Parameters to calculate enthalpy, entropy and Gibbs free energy for the Cu-Cr system.

	*G* (GPa)	*V* (10^−6^ m^3^/mol)	*K* (GPa)	Φ (V)	*n*^1/3^ (m^−1^)
Cu	48	7.1	137	4.45	147
Cr	115.3	7.12	160.2	4.65	173

**Table 2 materials-13-05532-t002:** Variations in the crystallize size, lattice parameter and dislocation density.

Milling Time (h)	0	2	4	8	10	12
Crystallize size (nm)	598.35 ± 4.6	235.35 ± 5.6	208.25 ± 3.2	156.45 ± 4.9	146.89 ± 2.1	133.54 ± 4.9
Lattice parameter (nm)	0.36156	0.36175	0.36178	0.36183	0.36184	0.36186
Dislocation density (×10^16^ m/m^3^)	-	6.46	8.60	10.6	12.0	13.0

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
