# Peer review of "Extension of Solid Solubility and Structural Evolution in Nano-Structured Cu-Cr Solid Solution Induced by High-Energy Milling"

_materials, 2020, doi:10.3390/ma13235532_

Round 1

Reviewer 1 Report

The authors of the article : Extension of Solid Solubility and Structural Evolution in Nano-structured Cu-Cr Solid Solution induced by High-energy Milling present few interesting results in a field with many articles in last 10 years. 

l95:(1) should be on line 94 and also for the rest of the equations , exept (7) 

L108-109: the authors must present the preparation technique for TEM analyze in order to understand the proccesses applied to samples  

Line227: a standard deviation (obtained from the 100 diameters determination) should be supply in text or on figure . 

Reviewer 2 Report

In this work, the authors present usage of the High Energy Ball Milling (HEBM) technology to extend the Cr solid solubility level in the Cu matrix. The study incorporates thermodynamics calculation to support and explain XRD analysis and microscopy (TEM) characterization of the product differentiate by the HEBM process.

The manuscript will be reconsidered appropriate for publication in Materials only after major revision. The authors should revise their manuscript according to the following remarks.

Line 94:

The authors employed the Scherrer formula to estimate the crystalline size. This technique is justified only if the corrected physical broadening was used (excluding bias instrumental broadening), and thorough consideration of standard reference material. I would ask the authors to better describe their practice in the manuscript.

On that account, I wonder why the authors did not use the Williamson-Hall technique (or modified W-H) to better determine the intrinsic strain.  Again, the authors should explain their practice in the manuscript.

Line 100:

The intrinsic strain is associated with point defects, grain boundary, triple junction, and stacking faults. This should be accounted for due to the SFE of Cu alloys. The authors should consider this as they attribute strain only to dislocations.

Line 166:

In their XRD analysis, micro-strain and domain size were estimated only from the broadening of the Cu (111).  For some reason, the authors assume uniform deformation, of isotropic nature. This assumption must be justified in the manuscript.

Line 172: Peak broadening should also be associated with strain and not strictly with solubility. This broadening is expected, particularly for the MA process which stands as SPD.

Line 184:

I would believe that this analysis is correct if the authors employed a method that considers instrumental and sample influence, such as the well-known Rietveld method or Le Bail (for example). Please address in the manuscript.

Line 187: I wonder why Vegard’s law was not used to better estimate the solubility of Cr solute atoms in Cu.  Please explain.

Reviewer 3 Report

The authors present a well-structured manuscript about the evolution of the Cu-5 wt% Cr solid solution by mechanical alloying. The experimental methods were conducted thoroughly and the discussion of the results was meticulous.

However, before accepting this research work for publication some details must be amended.

Introduction.

Line 4-6: Would be very helpful if the author incorporates more details of the researches described in reference 12-14 and 15. This could highlight the importance of your work.

There are sentences with fonts in gray color. Please changed.

Experimental

Line 81: Please incorporate the size and composition of the milling balls, as well as the milling container and total powder produced at each milling round.

There are a few descriptions of the High-energy milling procedure, it seems pretty much a standard method for MA. Please briefly describe why your method has to be considered as HEBM?

Results and discussion

  • Line 166: Must be Figure 3, not 2.
  • Table 2. Please remove the excess of decimal values.  Lattice parameter variation of 0.000001 nm is insignificant and hardly detectable.
  • Please take note of my comments written within your text in red.

Figure 6 displays the changes of Cr solubility and Cu internal strain with respect to milling time. With the increase of milling time, Cr solubility and the internal strain of Cu both increase rapidly, especially up to 8 h of milling time (I would say especially at the 2-4 h of milling for the Cr solubility). The increase of Cu internal strain is mainly due to the refinement of crystallize size, as displayed in Table 2. However, with MA treatment for more than 8  h, the increase trend of internal strain and solubility become a little slower, the same change trend as crystalize size and lattice parameter (Crystalize size decreased rapidly up to 4h of milling time, and slowly decreased from there. However, the lattice parameter increased in a steady-stage during the MA process). The slower increase rate after 8 h (in the case of the internal strain the increase is slow for the entire time of milling, while the Cr solubility increased sharply within the initial 4 h of milling)  would be attributed to the diffusion of Cr becoming difficult. Also, it can be noted that the Cr solubility (1.07 at.%, 1.15 at.%) reach to the usually saturation (0.89 at.%) with milling time of 10 h and 12 h (I would say at only 4 h of milling time). The supersaturated Cu-Cr solid solution powder is prepared successfully with MA treatment for more than 8 h in this work. According to the trace of Cr solubility curve, the Cr solubility tend to increase very slowly or reach to a constant with the increase in milling time, as well as the crystallize size.

To sum up: In my opinion, after the initial 4 h of milling Cr atoms are rapidly incorporated into the Cu crystal structure, further milling would achieve primarily a grain refinement effect.

  • There is no mention of the possible contamination of the milling balls and/or container. A single line or 2 must be incorporated.
  • Please add a metallographic optical image of your heat-treated particles. This result would give an idea of how much Cr was finally incorporate into the Cu phase, or demonstrated that the undissolved Cr remains in a nano (submicro)-size.

Round 2

Reviewer 2 Report

The authors corrected the manuscript following the previous revision. 

All the best

Author Response

Thank you very much for all your help and comments. We have uplaod the Minor Revisions. Thank you!

Reviewer 3 Report

Congratulations on your revision "Extension of Solid Solubility and Structural Evolution in Nano-structured Cu-Cr Solid Solution induced by High-energy Milling".

After this first revision, all major points raised were solved adequately. There are still some minor suggestions for your consideration (below) and, once you've considered these, I look forward to accepting your paper for publication.

Table 2. I still believe you should write the lattice parameter as "0.XXXX" without the +/- error

Author Response

Please see the attachment, thank you!
